# Sp-R-Ip: A Decision-Focused Learning Strategy for Linear Programs that Avoids Overfitting

## Abstract

For forecast-informed linear optimization problems, neural networks have shown to be effective tools for achieving robust out-of-sample performance. Various decision-focused learning paradigms have further refined those outcomes by integrating the downstream decision problem in the training pipeline. One of these strategies involves using a convex surrogate of the regret loss function to train the forecaster, called the SPO+ loss function. It allows for the training problem to be reformulated as a linear optimization program. However, this strategy has only been applied to linear forecasters, and is prone to overfitting. In this paper, we propose an extension of the SPO+ reformulation framework that solves the forecaster training procedure using an interior-point optimization method, and tracks the validation regret of intermediate results obtained for different weights of the barrier term. Additionally, we extend the reformulation framework to include the possibility of neural network forecasters with non-linear activation functions. On a real-life experiment of maximizing storage profits in a day-ahead electricity market using actual price data, we show that the proposed methodology effectively solves the problem of overfitting, and that it can outperform other decision-focused benchmarks including training the forecaster with implicit differentiation.

## 1 Introduction

Linear programs (LPs) are ubiquitous in modern-day decision-making problems in operations research and finance. For many applications, the primary challenge of the problem:

$$\begin{aligned} \underset{x}{\text{minimize}} \quad & \hat{c}^T x \\ \text{subject to} \quad & Ax \geq b, \end{aligned} \tag{1}$$

lies in representing the forecast $\hat{c}$ of ground truth $c$ as accurately as possible. This is the case for optimal scheduling of assets in energy markets, like storage systems, see e.g. Byrne et al. (2018); financial portfolio optimization, see e.g. Mansini et al. (2014); and deterministic inventory models, see e.g. Levi et al. (2004). The utilization of Machine Learning (ML) models has become common practice for modeling those forecasts. While the particular modeling strategy may differ among the various ML techniques, the main idea is always to fit a generic model's parameters such that its output matches the ground truth given a set of inputs. The problem of training the forecaster, also referred to as the Empirical Risk Minimization (ERM) problem, involves minimizing a certain loss function over a train data set, which is in many cases solved with an implementation of the gradient descent algorithm.

Within the above context, an aspect that is increasingly looked into is the loss function $\mathcal{L}$. The traditional approach is to apply a statistical error metric, e.g. Mean Squared Error (MSE), between the forecast and ground truth. This is a sensible approach when the downstream decision problem is unknown, resulting in a generic forecaster. In Decision-Focused Learning (DFL) or end-to-end learning, the downstream (optimziation) problem is included in the forecasting pipeline. This typically includes adopting a task-aware loss function like the regret loss. For many downstream problems, including (mixed integer-)linear optimization, it is a well-known difficulty to minimize regret

with a gradient descent procedure. This is because of ill-defined gradients of the optimized output w.r.t. the forecast cost. Smoothing terms and perturbations with random noise have been proposed to overcome the issue, but such approaches involve approximating the downstream problem in the training phase and can as such distort the results.

Another promising approach was proposed by Elmachtoub & Grigas (2022), involving a convex surrogate of the regret loss function, coined "SPO+" loss. They also show that the SPO+ loss allows for DFL with dowsntream LPs in two distinct ways. First, whereas the SPO+ loss is unsuitable for calculating exact gradients w.r.t. the forecasted cost, its convexity does allow for the subgradient method to be used for training the forecaster. Secondly, the training problem can be re-written to a single-level optimization program by applying duality theory. The latter technique is largely unexplored, and was only implemented for a linear forecasting model. In this paper, we build upon this idea by solving the single-level reformulation of the ERM with an Interior Point (IP) method. This approach, which is inspired by early stopping in traditional gradient descent training, allows for tracking the validation performance along the training iterations. Since IP methods can be used for non-convex optimization problems, this method also allows for extending the forecaster to a broader class of non-convex ML models, such as various types of Neural Networks (NN). The proposed approach comes at the cost of slower training compared to gradient-based methods. We mitigate this by adopting a two-stage procedure, where an initial large-scale forecaster is trained to minimize a statistical error metric. In the second stage, a smaller re-forecaster is trained by adopting the proposed DFL technique.

## 2 RELATED WORK AND CONTRIBUTIONS

**Decision-focused learning with gradient descent**

When solving DFL with gradient descent training, implicit differentiation of the KKT conditions can be used to compute gradients "through" an optimization program to update the parameter of the NN preceding the optimization. This line of research was catalyzed by the seminal work of Amos & Kolter (2017) and Donti et al. (2017), where such differentiation was achieved for quadratic programs. Agrawal et al. (2019) extended the method to convex optimization problems that can be written as disciplined parametric programs. The discontinuous nature of the output of linear and combinatorial programs is well-known to pose difficulties when trying to apply such methods. This sparked Wilder (2019) to add a quadratic smoothing term in the objective of the linear program, thereby enabling the calculation of gradients and facilitating the use of the above-mentioned methods. This idea was extended by Ferber et al. (2020) with a cutting planes approach to address mixed-integer linear programs as the downstream problem. In similar fashion and inspired by IP solution procedures, Mandi & Guns (2020) introduce a log-barrier smoothing term, rendering the objective function strictly convex. An approach that is conceptually similar to such smoothing terms is that of including a tunable noise term to the predicted cost, which was used in an additive, see Berthet et al. (2020), and multiplicative, see Dalle et al. (2022), fashion. For a comprehensive overview of DFL techniques, we refer the reader to Kotary et al. (2021) and Mandi et al. (2023).

**Interior point methods in machine learning**

Whereas gradient descent methods have dominated the field of ML recently because of their excellent scaling properties, other (albeit often slower) approaches of training ML models have been proposed, including IP solution strategies. This was first introduced by Trafalis et al. (1997) in a generic NN training setting. Koh et al. (2007) and Li & Liu (2022) show the effectiveness of IP-based solving of a logistic regression problem, which is widely used in the context of feature selection. Another sub-field of ML where IP methods gained significant traction is the training of Support Vector Machines (SVM). SVMs are mostly used for classification tasks, and their training problem can often be formulated as a quadratic program (QP). Ferris & Munson (2002), Woodsend & Gondzio (2011) and Gu et al. (2023) exploit the ability of interior point methods to efficiently solve for the global optimum of such QPs. It is noteworthy that none of these works discuss the possibility of tracking the validation performance along the iterations of the solution procedure, and rely solely on regularization terms to avoid overfitting.

**Contributions**

The scientific contribution of this work is threefold: (i) We develop an interior point-based neural network training algorithm that iteratively tracks the task-specific validation loss and dynamically updates the barrier term. (ii) We extend the original SPO+ reformulation ERM to accommodate neural network forecaster training. (iii) We demonstrate the effectiveness of the method by showcasing reduced out-of-sample regret values compared to state-of-the-art DFL models for an optimal scheduling problem of a storage system under price uncertainty using real-life data.

## 3  DECISION-FOCUSED LEARNING

When training a forecaster $f : \mathbb{R}^n \to \mathbb{R}^m$, mapping a vector of input features $\alpha \in \mathbb{R}^n$ to the forecast $\hat{c} \in \mathbb{R}^m$, the ERM problem can be written as:

$$\min_{\theta \in \Theta} \sum_{i \in \mathcal{I}_{tr}} \mathcal{L}(c_i, f(\alpha_i; \theta)) + \lambda \Omega(f), \tag{2}$$

with $\theta$ the trainable parameters of the forecaster, $\mathcal{I}_{tr}$ the indices in the train data set, $(\alpha_i, c_i)$ vector instances of the labeled trainset, $\mathcal{L}$ the chosen loss function and $\Omega$ a regularization function that helps in avoiding to overfit on the train data. In DFL, it is acknowledged that the forecast is deployed in a downstream decision problem. A typical loss function that is chosen for DFL is the regret loss. When the downstream decision problem exhibits a linear objective function, the regret is defined as:

$$r(c, \hat{c}) = c^T x^*(\hat{c}) - c^T x^*(c), \tag{3}$$

being the difference in the ex-post downstream objective value of the optimal decisions $x^*$ based on the forecasts, compared to that of the optimal decisions based on the ground truth. Setting the loss function in (2) to this regret function, the ERM becomes a bi-level optimization problem:

$$\min_{\theta \in \Theta} \left[ \sum_{i \in \mathcal{I}_{tr}} c_i^T \cdot \arg\min_{x \in \{x | Ax \geq b\}} \left( f(\alpha_i; \theta)^T x \right) \right], \tag{4}$$

which is well-known to be intractable for large-scale problems. In recent years, Neural Networks (NNs) have shown outstanding peformance in many machine learning applications. When using nonlinear activation functions in the NN architecture, the ERM problem becomes highly non-convex regardless of the loss function, and is typcially solved with gradient descent. Here, the NN parameters are iteratively updated by calculating the gradient of the loss function with respect to that parameter, and taking a step in the direction of steepest descent:

$$\theta \leftarrow \theta - \psi \frac{\partial \mathcal{L}}{\partial \theta}, \tag{5}$$

where $\psi$ represents the learning rate. When regret is used as the loss function in the ERM, the gradients can be calculated by using the chain rule:

$$\frac{\partial \mathcal{L}}{\partial \theta} = \frac{\partial r}{\partial x^*} \frac{\partial x^*}{\partial \hat{c}} \frac{\partial \hat{c}}{\partial \theta}. \tag{6}$$

The first and third factors can be straightforwardly calculated with traditional techniques. The second factor, $\frac{\partial x^*}{\partial \hat{c}}$, being the gradient "through" the optimization program, can be calculated with implicit differentiation of the KKT optimality conditions, see Amos & Kolter (2017). Considering a downstream optimization problem of the form $\min_{x \in S} g(x, \hat{c})$, this leads to the following set of equations:

$$\begin{bmatrix} \frac{\partial^2 g}{\partial x^* \partial \hat{c}}(x^*, \hat{c}) \\ 0 \end{bmatrix} + \begin{bmatrix} \frac{\partial^2 g}{\partial x^{*2}}(x^*, \hat{c}) & -A^T \\ A & 0 \end{bmatrix} \begin{bmatrix} \frac{\partial x^*}{\partial \hat{c}} \\ \frac{\partial \lambda}{\partial \hat{c}} \end{bmatrix} = \begin{bmatrix} 0 \\ 0 \end{bmatrix}. \tag{7}$$

In order to solve for $\frac{\partial x^*}{\partial \hat{c}}$, the Hessian of the objective function should be non-zero, which is not the case for a linear optimization problem. To overcome this, smoothing terms have been proposed

in the form of a quadratic term by Wilder (2019) and a log-barrier term by Mandi & Guns (2020). This indeed results in gradients that are readily computable. However, as the perturbed optimization problem is an approximation of the actual downstream problem, results may be suboptimal, as we show in Section 5.

**SPO+ reformulation**

Elmachtoub & Grigas (2022) propose an alternative approach and acknowledge that the regret function is non-convex in $\hat{c}$, which poses challenges in the training process. They re-write it to the "SPO+" loss: $l^{SPO+} = \max_{x \in S}\{c^T x - 2\hat{c}^T x\} + 2\hat{c}^T x^*(c) - z^*(c)$, and argue that when the underlying uncertainty distributions are well-behaved, minimizing this SPO+ loss corresponds to minimizing the regret function. In this formulation, $z^*(c)$ represents the optimal objective value. Interestingly, the SPO+ loss function is convex in $\hat{c}$ and the authors provide an expression for a subgradient:

$$2\left(x^*(c) - x^*(2\hat{c} - c)\right) \in \partial_{\hat{c}} l^{SPO+}(c, \hat{c}). \tag{8}$$

This presents the opportunity to train a neural network to minimize this surrogate of the regret loss function using the subgradient method. However, our results in Appendix D.2 demonstrate that this expression of the subgradient can lead to suboptimal NN parameter updates in the training procedure. Elmachtoub & Grigas (2022) also provide a second, reformulation, approach: by substituting $l^{SPO+}$ in (2), and leveraging duality theory, they find an ERM which is a linear optimization program when the forecaster is assumed to be a linear function of the input features, i.e. $\hat{c} = B\alpha$. The ERM is given by:

$$\begin{aligned}
\underset{B,P}{\text{minimize}} \quad & \sum_{i \in \mathcal{I}_{tr}} [-b^T p_i + 2(x^*(c_i)\alpha_i^T) \bullet B - z^*(c_i)] + \lambda|B| \\
\text{subject to} \quad & A^T p_i = 2B\alpha_i - c_i && \forall i \\
& p_i \geq 0 && \forall i,
\end{aligned} \tag{9}$$

where $P$ represents the set of vectors $p_i$, containing the dual variables associated with the constraints of the downstream optimization problem for every train sample $i$. In the objective value, $\bullet$ refers to the trace inner product. More details of this reformulation can be found in Appendix A. Thus, the bi-level ERM (4) is reformulated to a linear optimization program that can be solved with off-the-shelve solvers like CPLEX and Gurobi. No approximation of the mapping from forecasted parameter to decision was required. However, the limitation of this approach is threefold: (i) the training procedure has only been implemented for linear forecasters which restricts the predictive power of the forecaster, (ii) there is a significant risk of overfitting as the result gives the minimum objective value over the training set, and (iii) the ERM problem does not scale well, making it to a lesser extent applicable compared to a (sub)gradient-based approach when large amounts of data are required for the training procedure.

## 4 METHOD

We aim to enhance the SPO+ reformulation framework by addressing two of the above-mentioned key limitations. The first extension is presented in Section 4.1, where we introdue the Sp-R-IP training method. This is an iterative interior point solution procedure to address the problem of overfitting. In section 4.2, we provide a second extension which consists of accommodating non-linear NN forecasters in the SPO+ framework, improving the predictive power of the trained forecast. Finally, Section 4.3 outlines a re-forecasting procedure designed to mitigate the scalability limitation of the IP-based solution method.

### 4.1 SP-R-IP

In training machine learning models, and especially NNs, overfitting to the train data is a well-known problem. The two main procedures to avoid overfitting are (i) regularization and (ii) early stopping. When deploying a regularization term in the ERM problem, high values in the parameters of the

forecaster $f$ are penalized, which may lead to models which generalize better. Early stopping entails the procedure of tracking, with each iteration in the (gradient descent) parameter update procedure, the performance of the current value of $f$ on the validation set, i.e. data which is not included in calculating the gradient. When the forecaster stops improving the validation performance, the training procedure is terminated.

Inspired by early stopping with gradient descent methods, we here propose a validation performance tracking procedure for the SPO+ reformulation. This requires an iterative approach to solving ERM (9). We observe that a simplex method will always explore vertices of the feasible region in its solution procedure. On the other hand, interior point methods can also be used to solve linear optimization problems and are arguably more suitable for this methodology. Indeed, the explored feasible points in the interior of the feasible region, which correspond to varying manifestations of the forecaster in problem (9), intuitively yield forecasters that generalize better than those derived from the extreme points of the feasible region. A second reason for using this IP-based solution procedure is that it accomodates the extension of ERM (9) to include NN forecasters, which renders the ERM non-linear, and therefore simplex methods inapplicable. The proposed method computes the validation performance of all forecasters obtained from the intermediate solutions accessed by an IP-based solver, and selects the forecaster with the best performance to be deployed on the test set. This SPO+ - Reformulation - Interior Point (Sp-R-IP) method is the core of our contribution.

IP solution procedures involve the use of a barrier method as standard practice. Generalizing, we can write a non-linear optimization problem as:

$$
\begin{aligned}
\underset{x}{\text{minimize}} \quad & g(x) \\
\text{subject to} \quad & c(x) = 0, \\
& x \geq 0.
\end{aligned}
\tag{10}
$$

The inequality constraint is replaced with a log-barrier term in the objective function:

$$
\begin{aligned}
\underset{x}{\text{minimize}} \quad & g(x) - \mu ln(x) \\
\text{subject to} \quad & c(x) = 0,
\end{aligned}
\tag{11}
$$

with $\mu$ a positive number. The barrier term resulting from this nonzero $\mu$ penalizes solutions close to the boundary of the feasible region. As $\mu \to 0$, the barrier problem approaches the original one. The collection of optimal points for all possible values of $\mu$ is the central path: $CP = \{x^*(\mu)|\mu \in \mathbb{R}_0^+\}$. The essence of interior point solvers is to iteratively decrease the value of $\mu$ and approximately following the central path toward the optimal solution. To find a solution of problem (11) for a specific value of $\mu$, one could invoke the KKT conditions and apply a Newton-Raphson procedure to iteratively update primal and dual variables towards intermediate solutions for a specific weight of the barrier term. Commercial and open-source solvers generally prioritize speed of obtaining the final (locally) optimal solution and design specialized algorithms for obtaining steps in the primal and dual variable space, and obtaining updated values of $\mu$, for that purpose. However, we argue that when the optimization program is an ERM for training a forecaster, the points on the central path should be regarded as actual intermediate solutions to be tested on the validation set. Indeed, these points constitute different realizations of the forecaster's parameters, exhibiting decreasing regret as $\mu$ decreases and the optimal value is allowed closer to the edge of the feasible region. As such, the points on the central path are the direct equivalent of the intermediate realizations of a forecaster as it is updated with the gradient descent method. For that reason, we propose to prioritize obtaining optimal solutions of (11) for relevant values of $\mu$ over speed of getting to the optimal solution on the train set. To that end, we propose a dynamic update strategy of gradually decreasing $\mu$, adapting the rate of decrease based on the validation performance:

$$
\mu_{n+1} = \frac{\mu_n}{d \cdot Z_{1,n} \cdot Z_{2,n}},
\tag{12}
$$

with

$$
Z_{1,n} = \begin{cases} 1 - \epsilon_1, & \text{if } v_n < v_{n-1} \\ 1 & \text{otherwise} \end{cases}
\tag{13}
\qquad
Z_{2,n} = \begin{cases} 1 - \epsilon_2, & \text{if } v_n < v_i, \forall i < n \\ 1 & \text{otherwise.} \end{cases}
\tag{14}
$$

In Eq. (12), $d$ represents a constant rate at which $\mu$ is decreased. In Eqs. (13-14), $v_n$ denotes the validation performance, i.e. the value of the metric to be minimized, while $\epsilon_1$ and $\epsilon_2$ are small predetermined constants that modulate the rate at which $\mu$ decreases. Specifically, $\mu$ decreases more slowly when the latest validation performance improves compared to the iteration before (13) or sets a new best score (14). Adopting this dynamic update strategy ensures a more granular search in areas of high validation performance. Algorithm 1 depicts a high-level overview of the training procedure.

---

**Algorithm 1** Sp-R-IP algorithm

---

1: **Input:** $\mathcal{D}_{tr} = \{\alpha_i, c_i | i = 1, ..., N_{tr}\}, \mathcal{D}_{val} = \{\alpha_j, c_j | j = 1, ..., N_{val}\}$     ▷ train and validation data
2: **Initialize** $\mu$                                                            ▷ barrier weight
3: **Initialize** $p_0, d_0$                                         ▷ primal and dual variables
4: **for** $n = 1, ..., epochs$ **do**
5:     $p_n, d_n \leftarrow \text{SolveOpti}(p_{n-1}, d_{n-1}, \mu, \mathcal{D}_{tr})$             ▷ using Problem (23)
6:     **Retrieve** $\theta_n \in p_n$
7:     $v_n \leftarrow \text{ValPerfo}(f(\cdot; \theta_n), \mathcal{D}_{val})$
8:     **if** $v_n < \min(\{v_i | i = 1, ..., n - 1\})$ **then**
9:        bestNet $\leftarrow f(\cdot; \theta_n)$
10:    **end if**
11:    $\mu \leftarrow \text{updateMu}(\mu, \{v_i | i = 1, ...n\})$                  ▷ Via Eq. (12)
12: **end for**
13: **Output:** bestNet

---

## 4.2 NEURAL NETWORK IN SPO+ REFORMULATION

Existing implementations of the SPO+ reformulation ERM are limited to a linear forecaster. Here we propose to extend that framework to accomodate feedforward NN forecasters. Details of this derivation can be found in Appendix A. The ERM now reads:

$$
\begin{aligned}
\underset{W^{(l)}, b^{(l)}, P}{\text{minimize}} \quad & \sum_{i \in \mathcal{I}_{tr}} -b^T p_i + 2 \operatorname{Tr}(x^*(c_i)\hat{c}_i^T) + \lambda \sum_{(l)} |W^{(l)}| \\
\text{subject to} \quad & A^T p_i = 2\hat{c}_i - c_i & \forall i \\
& p_i \geq 0 & \forall i \\
& \alpha_i^{(l)} = a^{(l)}\left(W^{(l)}\alpha_i^{(l-1)} + b^{(l)}\right) & \forall i, (l) = 1, ..., L-1 \\
& \hat{c}_i = W^{(L)}\alpha_i^{(L-1)} + b^{(L)} & \forall i, \\
& c_{i,init} - \xi|c_{i,init}| \leq \hat{c}_i \leq c_{i,init} + \xi|c_{i,init}| & \forall i,
\end{aligned}
\tag{15}
$$

with $\operatorname{Tr}()$ the trace operator, $L - 1$ the total amount of NN hidden layers, $W^{(l)}$, $b^{(l)}$ and $a^{(l)}$ the weights, biases and (nonlinear) activation function of layer $(l)$ respectively, and $\alpha^{(0)}$ the input features. This extension of the SPO+ ERM problem increases the predictive power of the forecaster, while rendering the ERM a nonlinear optimization program. As such, we loose the guarantee of finding a global optimum, and no longer have the option of solving the problem with simplex methods. In light of the discussion in the previous section, these drawbacks are arguably acceptable as the globally optimal solution is expected to be overfitted to the training set, and the interior point method is the option of choice for retrieving intermediate solutions that generalize well to unseen data. The last set of inequalities inhibits the NN to produce outputs that are very different from some initial forecast, $\hat{c}_{init}$ (see Section 4.3). $\xi$ Becomes a new (nonnegative) hyperparameter of the problem, where $\xi \to \infty$ results in effectively removing the constraint, and $\xi \to 0$ forces the forecaster to produce the initial forecast, which may result in an infeasible optimization program, depending on the train data and the architecture of the neural network. The purpose of this constraint is to further reduce the ability of the NN to overfit on the train data.

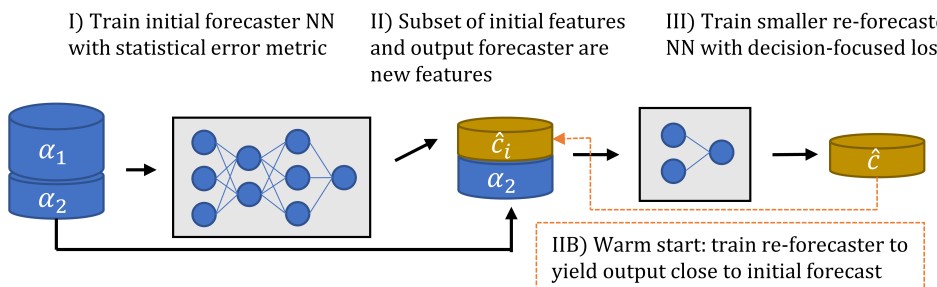

I) Train initial forecaster NN with statistical error metric

II) Subset of initial features and output forecaster are new features

III) Train smaller re-forecaster NN with decision-focused loss

IIB) Warm start: train re-forecaster to yield output close to initial forecast

Figure 1: Re-forecast method with optional warm start.

## 4.3 RE-FORECASTING AND MINI-BATCHES

As the proposed approach is a second-order method for training a NN, this is expected not to scale efficiently with the size of the forecaster or training dataset. here, two heuristic procedures are proposed to mitigate this problem, being a re-forecasting methodology and a mini-batch implementation.

**Re-forecasting**

To mitigate the problem of high computational time and memory usage, we adopt a two-stage approach. In the first stage, an initial forecaster is trained to minimize the MSE of cost forecasts on an initial train data set. In the second stage, a refined set of input features, comprising a subset of the original features and the output of the initial forecaster, are used to train a secondory NN. This "re-forecaster" is trained to minimize the decision-focused loss function - being either the SPO+ loss function or the regret loss - on an auxiliary data set. As such, the dimensionality of the decision-focused learning problem is strongly reduced, which improves the computational tractability. This methodology bears a resemblance to the concept of Large Language Model (LLM) adapters, see e.g. Hu et al. (2023). In this framework, a second (adapter) NN is trained on top of the LLM, which is not updated in the auxiliary task-specific adapter training procedure. To avoid that the re-forecaster NN gets stuck in local minima exhibiting performance worse than that obtained by the initial forecaster, we implement a warm start methodology. Here, before solving the ERM problem, the re-forecaster is trained to yield outputs that are close to the output of the initial forecaster (by minimizing the MSE). This pretrained re-forecaster is used as a starting point for the ERM problem. The 2-stage forecasting procedure with optional warm start is visualized in Figure 1.

**Mini-batches**

Another extension of the SPO+ reformulation framework constitutes the use of mini-batches, which can improve the scalability of the method, as exemplified in Appendix E. We here develop a naive approach where the training set is split up in $M$ mini-batches. The procedure outlined in Algorithm 1 is executed independently for all $M$ mini-batches independently. This entails training M distinct neural networks (NNs) and evaluating their performance on the complete validation dataset during the training process. While this leads to a loss of information for the separate NNs, this results in a greater variety of intermediate manifestations of the NN that potentially generalize better on unseen data.

## 5 EXPERIMENT

We apply our DFL training approach to the day-ahead scheduling problem of an Energy Storage System (ESS) in the Belgian electricity market, with the objective of maximizing the profit based on forecasts of the day-ahead electricity price. The details of the linear optimization problem governing the ESS decisions are given in Appendix B. We evaluate the performance of the proposed methodology through a comparative analysis involving eight distinct models. The baseline model relies solely on the initial forecaster's projections. The other methods deploy different DFL-oriented re-forecaster implementations on top of the initial forecast. We implement 2 methods using Implicit Differentiation (ID), one using the SPO+ loss function with SubGradient descent (Sp-SG),

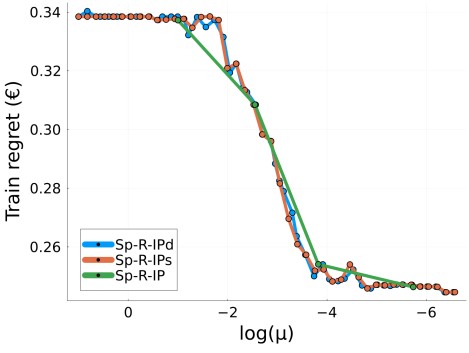
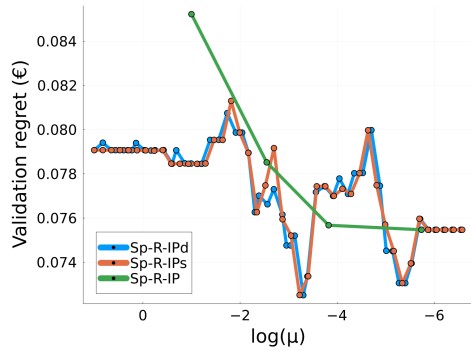

(a) Evolution of regret loss on train data set          (b) Evolution of regret loss on validation data set

Figure 2: Example of regret loss evolution on train (left) and validation (right) data throughout the training procedure for different IP-based training methods using a NN forecaster with a single hidden layer and softplus activation function.

the baseline SPO+ reformulation method (Sp-R) and three variants of our proposed methodology, collectively referred to as Sp-R-IPx. For all DFL models, both a linear regression (-lin) and a single hidden layer NN variant with softplus activation function (-softplus) are implemented. The different models have different choices of hyperparameters. More details on the models and hyperparameters are provided in Appendix C.

Figure 2 exemplifies the value of both the validation performance tracking procedure and controlling the barrier weight parameter. It displays in-sample training regret (Figure 2a) and out-of-sample validation regret (Figure 2b) of solutions to Problem (11) for decreasing values of the barrier weight $\mu$, comparing the Sp-R-IPx models. A first observation is that the results verify the viability of the SPO+ loss function. Indeed, the regret on the train set tends to decrease during the training procedure. Secondly, the figure serves as a compelling argument for the value of tracking the validation performance in the SPO+ reformulation training procedure. We remind the reader that the original reformulation approach, Sp-R, only considers the final optimal solution on the train set, represented by the (rightmost) point with the smallest value of $\mu$ in the figures. Whereas this in many cases corresponds to the optimal solution in terms of regret on the train set, Figure 2b) clearly shows that this is not necessarily the case for the validation set, leading to the conclusion that the Sp-R method is prone to overfitting. In contrast, the Sp-R-IPs/d models capture forecasters with improved validation performance for intermediate values of $\mu$ compared to the last accessed point. This underscores the importance of our proposed validation tracking procedure. Finally, the figure shows the significance of controlling the $\mu$ update in the solution procedure. Indeed, the Sp-R-IP model, which produces the intermediate results of the barrier problem via IPOPT, explores a limited range of $\mu$ values. As such, it fails to find the intermediate solutions in the regions $\log(\mu) \approx -3$ and $\log(\mu) \approx -5$, where the other models find solutions with comparatively lower validation regret. When examining the Sp-R-IPd and Sp-R-IPs methods, we observe a slightly lower obtained validation regret when dynamically updating $\mu$ based on the validation performance, compared to the static update. However, this is less significant than the improvement achieved by controlled $\mu$ updates over the IPOPT solver output.

Table 1 presents a comprehensive evaluation of the test performance across the different solution approaches. Absolute regret is calculated as the sum of the obtained regret over the days in test set, using standardized day-ahead prices. Relative regret measures this against the baseline of an ESS making decisions based on the output of the initial forecaster (inital FC in the table). As such, the relative regret is negative when the re-forecaster improves upon the performance of the initial forecaster. The first observation from the table is that the models from the proposed methodology (Sp-R-IPx) systematically outperform the models using a (sub)gradient descent method, and the original SPO+ reformulation approach, thus affirming the effectiveness of the proposed methodology. A second observation is that (sub)gradient-based models (ID-Q, ID-LB and Sp-SG) are unable to compete with the initial forecaster benchmark when the re-forecaster is cold-started. Deeper insights on these methods are provided in Appendix D. From those analyses, we conclude that the ID methods suffer from a difficult balancing exercise when choosing the value of the weight of the smoothing term,

Table 1: Out-of-sample regret obtained on the test set, for both absolute ($r_{abs}$) and the relative improvement ($r_{rel}$) compared to the regret obtained by the initial forecaster. "Time" refers to the total time elapsed during the train procedure.

| Model | Cold start | | | Warm start | | |
|---|---|---|---|---|---|---|
| | $r_{abs}$ (€) | $r_{rel}$ (%) | Time (s) | $r_{abs}$ (€) | $r_{rel}$ (%) | Time (s) |
| **Initial FC** | 0.095 | - | - | 0.095 | - | - |
| **ID-Q-lin** | 0.278 | +192 | 407 | 0.094 | -1.3 | 87 |
| **ID-Q-softplus** | 0.339 | +257 | 309 | 0.089 | -6.0 | 157 |
| **ID-LB-lin** | 0.162 | +70.5 | 2,031 | 0.094 | -1.3 | 1,960 |
| **ID-LB-softplus** | 0.191 | +101 | 1,309 | 0.084 | -11.6 | 1,970 |
| **Sp-SG-lin** | 0.270 | +184 | 2,410 | 0.101 | +6.4 | 2,320 |
| **Sp-SG-softplus** | 0.240 | +153 | 2,329 | 0.096 | +0.9 | 2,309 |
| **Sp-R-lin** | 0.100 | +5.1 | 340 | 0.100 | +5.1 | 738 |
| **Sp-R-softplus** | 0.086 | -8.9 | 2,051 | 0.090 | -5.4 | 1,314 |
| **Sp-R-IP-lin** | 0.082 | -14.0 | 338 | 0.082 | -13.6 | 738 |
| **Sp-R-IP-softplus** | 0.083 | -12.8 | 1,937 | 0.094 | -0.8 | 1,076 |
| **Sp-R-IPs-lin** | 0.082 | -13.9 | 1,606 | 0.080 | -15.7 | 1,295 |
| **Sp-R-IPs-softplus** | **0.079** | **-16.3** | 14,731 | **0.078** | **-17.2** | 6,075 |
| **Sp-R-IPd-lin** | 0.082 | -13.9 | 2,053 | 0.082 | -13.9 | 2,649 |
| **Sp-R-IPd-softplus** | 0.080 | -15.5 | 17,651 | 0.083 | -13.0 | 13,771 |

and that the expression of the subgradient provided in Elmachtoub & Grigas (2022) does not always provide optimal parameter updates in the training process. Thirdly, for all the models, the difference in regret between a linear and a NN re-forecaster with a single hidden layer is limited. Even though this result is context-specific and should not be generalized, it underpins the utility of our two-stage re-forecasting procedure where the initial forecaster captures complex dynamics, and the second-stage re-forecaster finetunes the output for improved downstream performance. The final observation is that the Sp-R-IPs/d methods tend to have longer train times compared to the (sub)gradient-based methods. Most notably, whereas the (sub)gradient-based methods show similar train times for the linear regression and NN re-forecaster architectures, the train time dramatically increases for the IP-based methods. This increase is necessitated by the transition from the linear ERM formulation (9) to its non-linear counterpart (23). Even so, the two-stage re-forecasting procedure ensured a tractable training procedure, yielding superior performance for the proposed methods relative to benchmarks, all within an acceptable time frame for the specific application.

## 6 CONCLUSION AND FUTURE WORK

While implicit differentiation methods have gained prominence in decision-focused learning for non-linear convex problems, their application to downstream linear optimization necessitates the inclusion of smoothing terms. Our findings indicate that such approximations may not yield optimal results. On the other hand, we show that the SPO+ reformulation framework is prone to overfitting. To address this issue, we augment the SPO+ reformulation methodology to incorporate validation performance tracking across training iterations, employing an interior point solver, while also extending the method to include neural network forecasters. We have shown that this approach outperforms available decision-focused benchmarks for the optimal scheduling problem of an energy storage system. However, this comes at an increased computational cost. While our proposed mini-batch approach seems to alleviate this problem to some extent, future research could investigate how this can be implemented in a more rigorous way.

## 7    REPRODUCIBILITY STATEMENT

In order to reproduce the results of this paper, readers can access the source code in the anonymous Github repository via this link: https://anonymous.4open.science/r/Sp-R-IP-55D2. This repository includes the data and scripts used for training the forecasters, as well as scripts for reproducing the figures.

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

## A    SPO+ OPTIMIZATION FRAMEWORK

Here, we elaborate on the optimization programs of the SPO+ reformulation framework. For completeness, we give the derivation of the original SPO+ reformulation as proposed by Elmachtoub and Grigas Elmachtoub & Grigas (2022). There, the authors start from a general ERM problem given by:

$$\min_{\theta \in \Theta} \sum_{i \in \mathcal{I}_{tr}} \mathcal{L}(c_i, f(\alpha_i; \theta)) + \lambda \Omega(f), \tag{16}$$

where a loss function $\mathcal{L}$ is minimized over a the training set $\mathcal{I}_{tr}$ by tuning the parameters $\theta$ of the forecaster $f$, potentially considering a regularizer $\Omega$. In the paper, a convex surrogate of the regret loss function (in the context of a linear downstream optimization program) is proposed, coined the SPO+ loss: $l^{SPO+}(c, \hat{c}) = \max_{x \in S}\{c^T x - 2\hat{c}^T x\} + 2\hat{c}^T x^*(c) - z^*(c)$. When considering a linear forecaster, i.e. $\hat{c} = B\alpha$, we can re-write (16) as:

$$\begin{aligned}
\underset{B}{\text{minimize}} \quad & \sum_{i \in \mathcal{I}_{tr}} [o_i + 2(x^*(c_i)\alpha_i^T) \bullet B - z^*(c_i)] + \lambda\Omega(B) \\
\text{subject to} \quad & o_i = \max_x \{(c_i - 2B\alpha_i)^T x \text{ s.t. } Ax \geq b\} \qquad \forall i.
\end{aligned} \tag{17}$$

This nested optimization program requires evaluating the objective value of the inner optimization which is a linear program. For linear programs in the form that appears in the lower level, strong duality dictates:

$$\begin{aligned}
\max_x \quad & cx & = \quad \min_p \quad & -b^T p \\
\text{s.t.} \quad & Ax \geq b & \text{s.t.} \quad & -A^T p = c \\
& & & p \geq 0,
\end{aligned} \tag{18}$$

where $p$ is the vector of dual variables associated with the constraints of the primal problem. This is used to re-write (17) to:

$$\begin{aligned}
\underset{B}{\text{minimize}} \quad & \sum_{i \in \mathcal{I}_{tr}} [o_i + 2(x^*(c_i)\alpha_i^T) \bullet B - z^*(c_i)] + \Omega(B) \\
\text{subject to} \quad & o_i = \min_p \{-b^T p \text{ s.t. } -A^T p = c_i - 2B\alpha_i, p \geq 0\} \quad \forall i,
\end{aligned} \tag{19}$$

which is equivalent to:

$$
\begin{aligned}
\underset{B,P}{\text{minimize}} \quad & \sum_{i \in \mathcal{I}_{tr}} [-b^T p_i + 2(x^*(c_i)\alpha_i^T) \bullet B - z^*(c_i)] + \Omega(B) \\
\text{subject to} \quad & A^T p_i = 2B\alpha_i - c_i && \forall i \\
& p_i \geq 0 && \forall i.
\end{aligned}
\tag{20}
$$

This is the same as problem (9) when using $L1$ regularization. Notice that the resulting optimization problem would be bi-linear (exhibiting a term $(B\alpha)^T x$) when strong duality were not used. Additionally, it is important to notice that the form of the SPO+ loss function is crucial for such a single-level reformulation to be possible: whereas the regret loss function exhibits the optimal decision, i.e. the argument minimizing the objective function of the downstream linear program, the SPO+ loss function includes the optimal value of the objective function.

Here, we show that we can apply the reasoning more broadly. When assuming no specific form of the forecaster, we can generalize ERM (17) to:

$$
\begin{aligned}
\underset{f \in \mathcal{H}}{\text{minimize}} \quad & \sum_{i \in \mathcal{I}_{tr}} [o_i + 2\hat{c}_i^T x^*(c_i)] + \Omega(f) \\
\text{subject to} \quad & o_i = \max_x \{(c_i - 2\hat{c}_i)x \text{ s.t. } Ax \leq b\} \quad \forall i \\
& \hat{c}_i = f(\alpha_i),
\end{aligned}
\tag{21}
$$

where $f$ can be any (non-convex) function from hypothesis class $\mathcal{H}$. Notice that we omitted $z^*(c_i)$ from the objective as this constant term does not affect the optimal values of the variables. The cost forecast $\hat{c}$ is a parameter in the inner optimization program. As such, the inner problem is still a linear optimization program to which we can apply strong duality. Hence, the ERM becomes:

$$
\begin{aligned}
\underset{f \in \mathcal{H}}{\text{minimize}} \quad & \sum_{i \in \mathcal{I}_{tr}} [o_i + 2\hat{c}_i^T x^*(c_i)] + \Omega(f) \\
\text{subject to} \quad & o_i = \min_p \{-b^T p \text{ s.t. } -A^T p = c_i - 2\hat{c}_i, p \geq 0\} \quad \forall i \\
& \hat{c}_i = f(\alpha_i),
\end{aligned}
\tag{22}
$$

Again, we can reformulate this to an equivalent single-level optimization problem:

$$
\begin{aligned}
\underset{f \in \mathcal{H},P}{\text{minimize}} \quad & \sum_{i \in \mathcal{I}_{tr}} -b^T p_i + 2\,\mathrm{Tr}(x^*(c_i)\hat{c}_i^T) + \Omega(f) \\
\text{subject to} \quad & A^T p_i = 2\hat{c}_i - c_i && \forall i \\
& p_i \geq 0 && \forall i \\
& \hat{c}_i = f(\alpha_i)
\end{aligned}
\tag{23}
$$

By replacing $f$ with constraints that correspond to the forward pass of a NN, we directly arrive at ERM problem (23).

## B    ESS PROFIT MAXIMIZATION

Here, we give details of the linear optimization program that maximizes the expected profit of the ESS based on price forecasts. The optimization problem reads:

$$\begin{aligned}
\underset{e^+,e^-,SoC}{\text{maximize}} \quad & \sum_{\tau=1}^{T} \hat{\lambda}_\tau^{DA} \left( e_\tau^+ \eta^+ - \frac{e_\tau^-}{\eta-} \right) \\
\text{subject to} \quad & 0 \le e_\tau^+ + e_\tau^- \le \overline{P}\Delta t && \forall \tau \\
& SoC_{\tau+1} = SoC_\tau + e_\tau^- - e_\tau^+ && \forall \tau \\
& \underline{\text{SoC}} \le SoC_\tau \le \overline{SoC} && \forall \tau \\
& SoC_0 = SoC_T = (\overline{SoC} - \underline{SoC})/2.
\end{aligned} \tag{24}$$

In the Belgian energy system, the day-ahead market is cleared once per day with a granularity of $\Delta t = 1h$. This results in a lookahead time horizon of $T = 24$. $\hat{\lambda}^{DA}$ represents the vector of forecasted day-ahead prices over the day. $e^+$ and $e^-$ represent the vectors of energy discharged from and charged to the battery, with discharge and charge efficiency $\eta^+ = \eta^- = 0.95$. The ESS is assumed to have maximum (dis)charge power of $\overline{P} = 0.01$ MW, and the maximum state of charge $\overline{SoC} = 0.04$ MWh, while the minimum state of charge $\underline{SoC} = 0$ MWh. With the cyclic boundary condition, the ESS avoids depleting its state of charge at the end of the day, as such maintaining favorable initial conditions for the subsequent day. This optimization program is straightforwardly recast to the form expressed in (1) by changing the sign of $\hat{\lambda}^{DA}$ and replacing the equality equations with two inequalities.

The initial forecaster adopts the "NBEATSx" architecture developed in Olivares et al. (2023) which has has demonstrated efficacy in electricity price forecasting. The feature set for this model comprises electricity generation forecasts both for Belgium and its neighbouring countries, load forecasts, and temporal variables. This data is publicly available. The model is trained on a historical data set spanning 2019 to 2022. The re-forecaster is trained with train data from January and February 2023. For the months of March and April 2023, the days are assigned to the validation and test set in alternating order. The main metric for evaluating performance is the regret loss, i.e. the difference in profit obtained by the ESS using a perfect price foresight, and that obtained by optimizing the schedule based on the price forecast. The prices used for calculating those profits were scaled with the standard deviation of the day-ahead price over the period January 2023 - April 2023.

## C    MODELS, HYPERPARAMETERS AND VALIDATION PERFORMANCE

Eight distinct decision-focused re-forecaster methods were tested. The first method (ID-Q) is one that implements Implicit Differentiation (ID) with a quadratic smoothing term, following Wilder (2019). The ID-LB method employs the same technique but considers a log-barrier smoothing term, inspired by Mandi & Guns (2020). The Sp-SG and Sp-R methods minimize the SPO+ loss, adopting the subgradient and original reformulation approaches respectively, both introduced in Elmachtoub & Grigas (2022). The final three models - collectively referred to as Sp-R-IPx - follow the IP-based method proposed in this paper. The first, Sp-R-IP, solves the SPO+ reformulation using the open-source IPOPT solver algorithm. In the second method, Sp-R-IPs, we intervene in the IP solution procedure by updating the barrier statically - i.e. $d = 1.5$, $\epsilon_1 = \epsilon_2 = 0$ in Eqs. (12-14). The third method, Sp-R-IPd, updates the barrier term dynamically with $d = 1.5$, $\epsilon_1 = \epsilon_2 = 0.1$.

Table 2 presents an overview of the hyperparameters that were varied in training the different models, and the values that were explored by means of a grid search in the hyperparameter tuning. For the ID models, the hyperparameter $\gamma$ refers to the weight of the smoothing term in the modified objective function, see Eq. (26). The hyperaparameter $\zeta$ refers to an additional constraint in Problem (9) that sets all NN weights to 0 which would allow passing information between different timesteps. That is, input features belonging to timestep $i$ cannot have an influence on $\hat{c}_j$ for $i \ne j$. Surprisingly, the results show that models with this restricted forecaster tend to outperform their non-restricted counterpart. The hyperparameters $lr$ and $bs$ refer to the learning rate and batch size respectively.

For completeness, Table 3 reports on the validation regret obtained by the different models. While the cold-started models of models ID-Q and Sp-SG here too yield unsatisfactory results, the Sp-SG method shows its ability to improve in terms of validation regret compared to the initial forecaster, in contrast to the test results shown in Table 1. Nevertheless, the key takeaway from the test results still holds: our proposed Sp-R-IPx methods generally outperform both (sub)gradient methods and the original Sp-R reformulation.

Table 2: List of hyperparameters per model and explored values.

| HP $\rightarrow$ | $\lambda$ | $\xi$ | $\zeta$ | $lr$ | $bs$ | $\gamma$ |
|---|---|---|---|---|---|---|
| **ID-Q** | $[0, 0.01]$ | - | - | $[5 \cdot 10^{-6}, 5 \cdot 10^{-5}]$ | $[8, 64]$ | $[0.1, 0.3, 1, 3, 10]$ |
| **ID-LB** | $[0, 0.01]$ | - | - | $[5 \cdot 10^{-6}, 5 \cdot 10^{-5}]$ | $[8, 64]$ | $[0.001, 0.003, 0.01, 0.03, 0.1]$ |
| **Sp-SG** | $[0, 0.0001, 0.01, 1]$ | - | - | $[0.001, 0.01, 0.1, 1]$ | $[16, 64]$ | - |
| **Sp-R** **Sp-R-IPx** | $[0, 0.001, 0.1, 10]$ | $[0.02, 0.1, 0.5, \infty]$ | $[0, 1]$ | - | $[16, 64]$ | - |

Table 3: Out-of-sample regret obtained on the validation set, for both absolute ($r_{abs}$) and the relative improvement ($r_{rel}$) compared to the regret obtained by the initial forecaster. Time refers to the total time elapsed during the train procedure.

| Model | Cold start | | | Warm start | | |
|---|---|---|---|---|---|---|
| | $r_{abs}$ (€) | $r_{rel}$ (%) | Time (s) | $r_{abs}$ (€) | $r_{rel}$ (%) | Time (s) |
| **Initial FC** | 0.079 | - | - | 0.079 | - | - |
| **ID-Q-lin** | 0.263 | +232 | 407 | 0.074 | -6.2 | 87 |
| **ID-Q-softplus** | 0.318 | +302 | 309 | 0.073 | -7.1 | 157 |
| **ID-LB-lin** | 0.148 | +87.3 | 2,031 | 0.074 | -6.3 | 1,960 |
| **ID-LB-softplus** | 0.154 | +94.9 | 1,309 | 0.072 | -8.6 | 1,970 |
| **Sp-SG-lin** | 0.267 | +238 | 2,410 | 0.073 | -7.1 | 2,320 |
| **Sp-SG-softplus** | 0.215 | +172 | 2,329 | 0.076 | -3.9 | 2,295 |
| **Sp-R-lin** | 0.073 | -7.6 | 340 | 0.073 | -7.6 | 738 |
| **Sp-R-softplus** | 0.071 | -9.9 | 2,051 | 0.075 | -4.7 | 1,314 |
| **Sp-R-IP-lin** | 0.070 | -12.0 | 338 | 0.072 | -9.3 | 738 |
| **Sp-R-IP-softplus** | 0.070 | -11.3 | 1,937 | 0.075 | -4.5 | 1,076 |
| **Sp-R-IPs-lin** | 0.069 | -12.6 | 1,606 | **0.067** | **-14.7** | 1,295 |
| **Sp-R-IPs-softplus** | 0.068 | -14.0 | 14,731 | 0.070 | -13.6 | 6,075 |
| **Sp-R-IPd-lin** | 0.069 | -12.6 | 2,053 | 0.069 | -12.6 | 2,649 |
| **Sp-R-IPd-softplus** | **0.068** | **-14.0** | 17,651 | 0.069 | -13.0 | 13,771 |

# D    RESULTS OF (SUB)GRADIENT-BASED METHODS

## D.1    ID-Q: IMPLICIT DIFFERENTIATION WITH QUADRATIC SMOOTHING

A widely-used strategy to smoothen downstream optimization problems exhibiting output decisions that are non-differentiable or exhibit zero gradients w.r.t. the input forecast is adding a term to the objective function of the original optimization problem. Wilder (2019) proposes a quadratic term for this purpose. In our case, the downstream linear problem (1) is modified to:

$$
\begin{aligned}
\underset{x}{\text{minimize}} \quad & \hat{c}^T x + \gamma x^T x \\
\text{subject to} \quad & Ax \geq b.
\end{aligned}
\tag{25}
$$

.

This is the optimization program that was used to obtain the results of the ID-Q method in Table 1. Alternatively, Mandi & Guns (2020) proposed to use a log-barrier term for the inequality equations to smoothen the output of the optimization program, as such mimicking an interior-point solution procedure. In this setting, the optimization program is re-written to:

$$
\begin{aligned}
\underset{x}{\text{minimize}} \quad & \hat{c}^T x - \gamma \sum_i \left( \ln(x_i) \right) \\
\text{subject to} \quad & Ax = b \\
& x \geq 0
\end{aligned}
\tag{26}
$$

,

which is the optimization program that was used to obtain the results for the ID-LB method. Both models were implemented using the Cvxpylayers Python library, developed by Agrawal et al. (2019).

Figure 3 shows the regret performance of an ESS when deploying the ID-Q forecaster in the original (left) and modified (right) optimization program for different values of the weight $\gamma$ of the smoothing term throughout the iterations of the training procedure. An SP-R-IPd benchmark is provided. All showcased models are trained without a regularization term. A number of insights follow from the figure. Firstly, for small values of $\gamma$, the NN is unable to improve the in-sample train profit, both for the original, as well as for the modified optimization problem. This can be explained by inspecting the amount of parameters in the NN experiencing a gradient of value zero at every iteration. We observe in the results that this number is highly correlated with the value of $\gamma$: a problem that is more strongly modified will exhibit less parameters without gradients in the training procedure. Consequently, for small values of $\gamma$, the gradient descent algorithm struggles to find the correct direction to update the parameters to. Secondly, for large values of $\gamma$, e.g. $\gamma = 30$, the train regret using the modified problem is strongly improved, whereas the regret obtained by using the original problem increases throughout the training iterations. This results from the smoothing term having too much weight, causing a mismatch between the actual objective of the ESS and the modified objective as seen by the gradient descent algorithm. Finally, none of the ID forecasters can match the in-sample performance of that obtained by the SP-R-IPd model. This can be attributed to two possible causes: (i) the gradients are always imperfectly calculated, as a balance must be struck between a smooth mapping from price forecast to optimal decision leading to well-defined gradients (large $\gamma$), and a correspondence between the original and modified problem (small $\gamma$); (ii) a gradient-descent algorithm is more prone to getting stuck in a locally optimal solution.

## D.2 SP-SG: SPO+ MINIMIZATION WITH SUBGRADIENT METHOD

In their paper introducing the SPO+ loss function, Elmachtoub & Grigas (2022) show that the SPO+ loss function is convex in $\hat{c}$, and that $2 \left( x^*(c) - x^*(2\hat{c} - c) \right)$ is a subgradient of $l^{SPO+}$ w.r.t. $\hat{c}$. When training a NN with the SPO+ loss, the chain rule can be used for subsequently calculating a subgradient of the SPO+ loss w.r.t. the NN parameter $\theta$, $\partial_\theta l^{SPO+}$. The training procedure outlined in Appendix C of the abovementioned paper and was implemented for the Sp-SG models in this work. Specifically, as update rule we adopted:

$$
\theta_{n+1} = \theta_n - \psi_n \partial_\theta l^{SPO+},
\tag{27}
$$

where the update step length is calculated following $\psi_n = lr / \sqrt{n+1}$, with $lr$ a pre-determined fixed learning rate, see Table 2. In Figure 4, we illustrate a specific situation that demonstrates how this subgradient can lead to suboptimal parameter updates. There, we consider a setting where a single-layer softplus NN is trained to minimize the SPO+ loss function on a train data set that contains a single day, February 28 2023. Figure 4a showcases that the NN reaches zero train regret after only two epochs. Whereas in an ideal situation, this would lead to zero gradient pressure for subsequent parameter updates, it is also visible that around epoch 120, the train regret surprisingly increases to a non-zero value. Figure 4b depicts forecasted the cost $\hat{c}$, i.e. the opposite of the price, as well as the ground truth and the cost difference appearing in the subgradient calculation for that day at epoch 100. Figure 4c shows the corresponding state of charge (i.e. optimal schedule) of the ESS throughout the day. Although $x^*(\hat{c}) = x^*(c)$, and hence the regret is indeed zero at epoch 100, it is demonstrated that $x^*(c) \neq x^*(2\hat{c} - c)$. From Equation (8), it follows that the subgradient is nonzero, being suboptimal as the optimal state was already achieved.

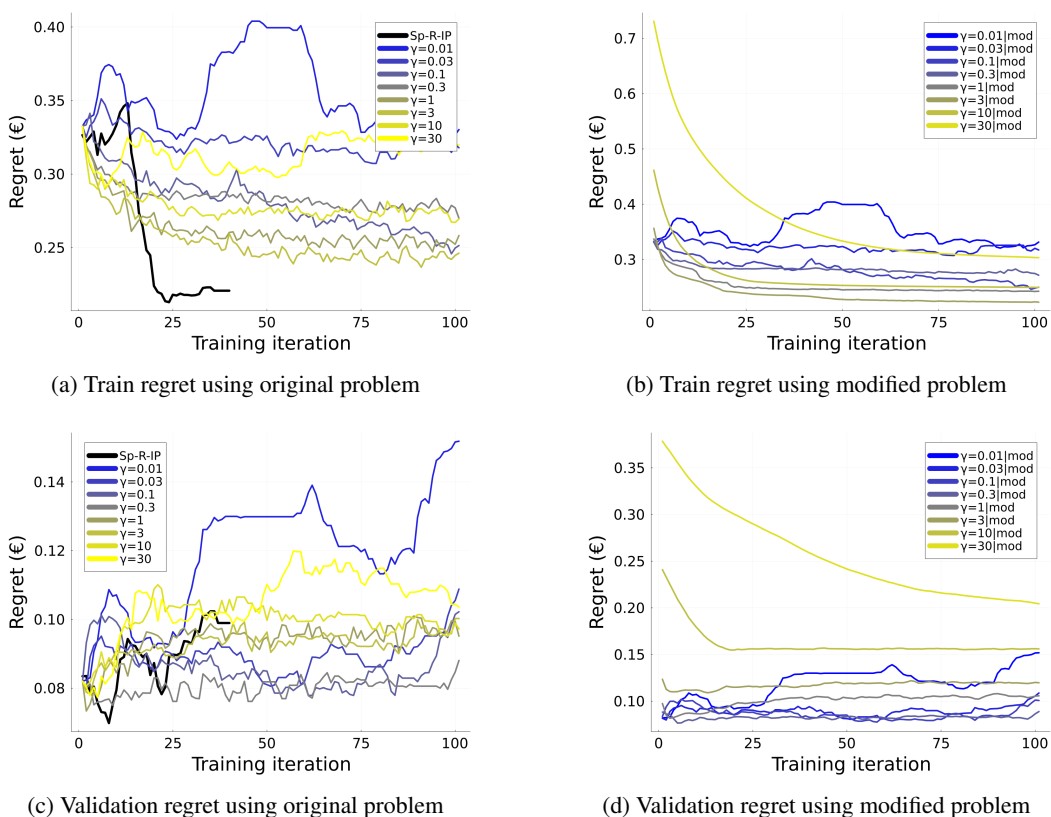

Figure 3: Evolution of train (top) and validation (bottom) regret obtained by an ESS making decisions using the original (left) and modified (right) problem throughout the training procedure of ID-Q models with softplus activation function . The degree of problem modification varies for the different models and is captured by the parameter $\gamma$ in Eq. (26). For the original problem, an Sp-R-IP benchmark is provided.

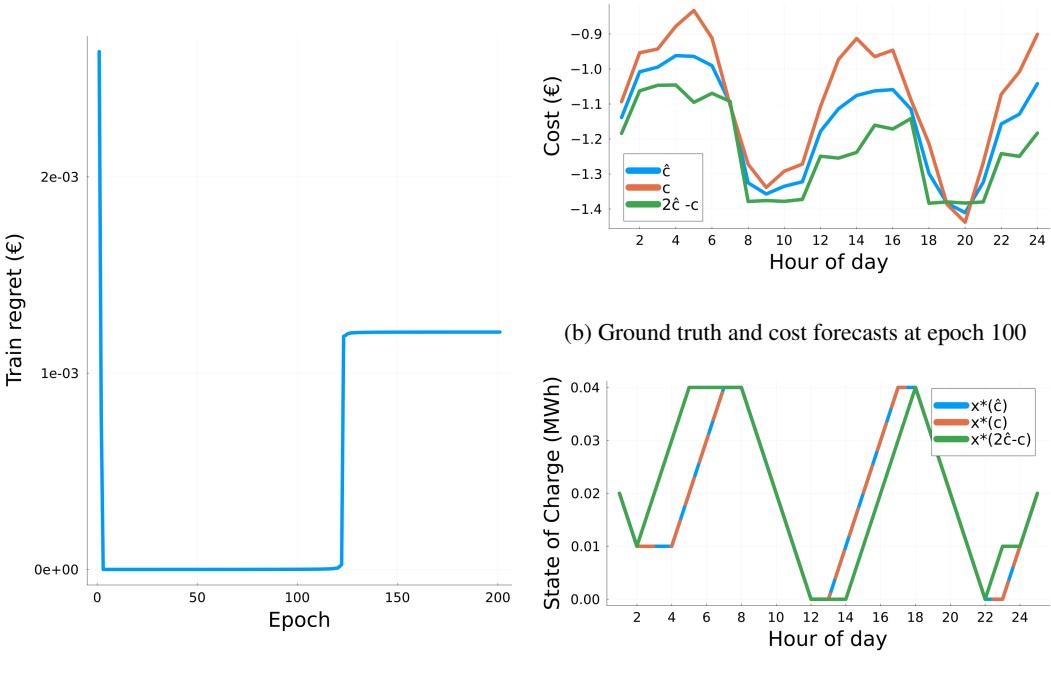

(b) Ground truth and cost forecasts at epoch 100

(a) Evolution of train regret over training epochs      (c) Optimal schedules at epoch 100

Figure 4: Evolution of train regret (left) and in-depth evaluation (right) of a forecaster with a softplus activation function trained using the Sp-SG method. The train set contains the single day of February 28 2023. On the right, we display the actual and forecasted costs, as well as the cost-like epxression appearing in the subgradient calculation (top), and their assoiated optimal schedules (bottom) at the 100th epoch.

## E  MINI-BATCH RESULTS

In this section, we showcase the impact of implementing the proposed mini-batch approach discussed in Section 4.3. We performed a sensitivity analysis of the results by running the experiment for different values of the batch size considering the Sp-R-IPs method, with one specific hyperparameter combination including a warm start. For the different batch size configurations, the experiment is run 10 times with different random choices of the specific batches. the outcome is shown in Figure 5. The left figure shows the average time required for training the models. The blue and orange bar show the total time for optimization and validation respecitvely, when the mini-batches are optimized in a sequential manner. The green bar shows the optimization time per mini-batch, which increases faster than linearly from batch size 16 onward. For this reason, the total time when training the mini-batches sequentially reaches a minimum around this point. Although the gains in time required to train are marginal this way, there is potential for siginficant reductions in train time when the mini-batches are optimized parallelly. The right figure shows the performance of the trained models in terms of regret on the validation set. Since the mini-batches are chosen randomly, the configurations with batch size smaller than the complete train set exhibit a distribution in performance over the ten runs, as opposed to the models trained with batch size 64. What can be concluded from this figure is that the naive mini-batch approach seems to perform similarly to the model trained with a single batch in terms of average regret performance. Based on these results, we chose the alternative batch size 16 for training the models Sp-R models.

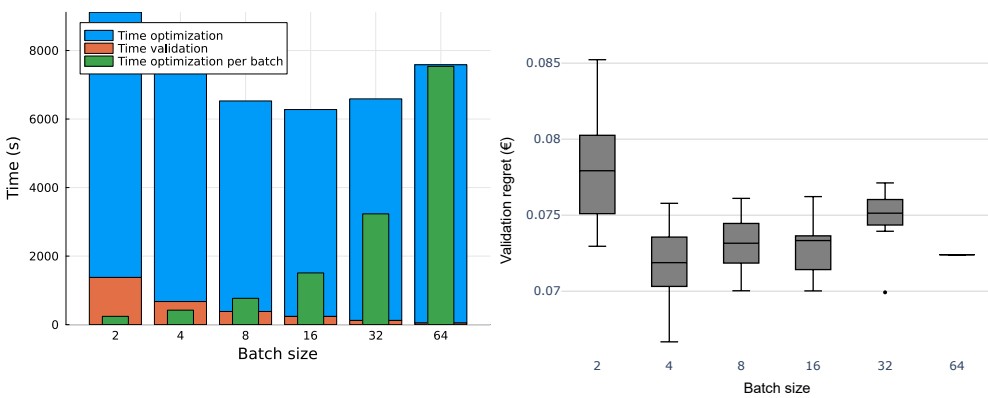

(a) Time of optimization and validation      (b) Regret performance on validation set

Figure 5: Sensitivity of average train time (left) and distribution of regret performance (right) of an Sp-R-IPs model with regards to the size of mini-batches, considering ten different random initializations of the batches.

