# OpenReview forum: "Sp-R-IP: A Decision-Focused Learning Strategy for Linear Programs that Avoids Overfitting"
_ICLR.cc/2024/Conference — Submitted to ICLR 2024_

### Official Review · Reviewer_AP5r · 2023-10-30

**Soundness:** 3 good
**Presentation:** 3 good
**Contribution:** 2 fair
**Rating:** 5
**Confidence:** 3

**Summary:**

The paper proposes an extension to the SPO+ reformulation of the decision-focused learning problem where the downstream decision problem can be framed with a linear objective. The SPO+ reformulation turns the decision-focused learning problem which is a bi-level optimization problem into solving a linear programming problem. The paper makes extensions on top of the SPO+ reformulation by introducing a neural network to forecast training and using the interior point method to solve the extended problem. To demonstrate the usefulness of the proposed method, the paper carried out experiments on a day-ahead energy scheduling dataset, comparing the proposed method with a wide variety of alternative methods.

**Strengths:**

* originality: the paper extends the SPO+ reformation of the decision-focused learning problem. I think the extension is not trivial but still somewhat incremental.
* quality: I think the paper properly identifies the limitation of the SPO+ reformation in using a linear forecaster. The paper also provides a solid solution to mitigate this limitation. The proposed method is compared to a variety of alternatives in the experiments.
* clarity: The presentation is clear. I can follow the paper.
* significance: It may help improve the effectiveness of solving the decision-focused learning problem via SPO+ reformulation, although I find it to be an improvement in a niche area.

**Weaknesses:**

* I find the paper's contribution to be somewhat incremental. Although I think the technical treatment described in the paper is non-trivial, the idea of replacing a linear forecaster with a non-linear one and the deployment of the interior point method appears to be straightforward observations.
* While the paper compared the proposed method with other competing methods, experiments are conducted on one dataset. This may suggest that the proposed method solves a problem in a niche area.
* The method appears to not be fairly efficient and scalable, as evidenced by the need for forecasting and warm starts.

**Questions:**

typo: " a second, reforumation, approach"
how is x^* compted?

---

> ### Author Response · Authors · 2023-11-23
> **Response to weaknesses and questions**
>
> Thank you for the time spent on this review and the feedback. Below you can find our response to the listed weaknesses and question, which we hope can convince you of the added value of the paper:
>
> W1: Although the proposed method builds upon the SPO+ reformulation framework as proposed by Elmachtoub & Grigas (2022), we believe that the use of a tailored interior point method to derive intermediate results that may generalize better on unseen data is a non-trivial step. Indeed, this contrasts with the prevailing emphasis in the optimization community on optimizing for speed in finding an optimum solution. Additionally, the extension to non-linear forecasters and the mini-batch approach that was added in the revised version of the manuscript substantially enhance the versatility of this reformulation approach.
>
> W2: See remark on ‘Amount of experiments’ in our comment to all reviewers.
>
> W3: See remark on ‘Scalability’ in our comment to all reviewers.
>
> Q1: Do you mean in equation (9) how you should compute $x^*(c_i)$? Since $c_i$ is the ground truth, this is pre-computed before optimization: $x^*(c_i)$ constitutes the optimal decision in the case of perfect foresight.

---

### Official Review · Reviewer_G7zN · 2023-10-31

**Soundness:** 3 good
**Presentation:** 2 fair
**Contribution:** 2 fair
**Rating:** 3
**Confidence:** 4

**Summary:**

This paper builds upon the dual formulation of the Linear Program (LP) Decision Focused Learning (DFL) framework, as initially introduced in Elmachtoub & Grigas (2022), in two notable ways:

1. The introduction of an interior point method to tackle the constraint optimization problem, enhancing the computational efficiency and accuracy of the framework.

2. Expansion of the framework to accommodate non-linear and non-convex mappings, such as neural networks, as opposed to the original work, which exclusively considered linear mappings.

In an empirical assessment conducted on a single dataset, the paper substantiates the effectiveness of the proposed approach.

**Strengths:**

- The paper presents a commendable mathematical formulation in Section 4, characterized by its clarity and logical coherence. This robust formulation lays a solid foundation for the subsequent analyses and conclusions.

- The experimental comparison conducted in this study is particularly commendable for its reliance on real-world practical data. This empirical approach not only enhances the relevance and applicability of the findings but also underscores the potential real-world impact of the proposed methodology.

**Weaknesses:**

- **Novelty and contribution**: The method outlined in this paper is a direct extension of the dual linear programming (LP) approach put forth in Elmachtoub & Grigas (2022). The first contribution, introducing the application of an interior point method (IP) for constrained optimization, represents a straightforward but meaningful addition from an optimization standpoint. Similarly, the second contribution involving the incorporation of neural network (NN) layers into constraints is conceptually clear-cut.

- **Scalability**: The optimization problem defined in equation 15 encompasses all NN layers and parameters, potentially leading to scalability challenges. The authors acknowledge this concern in Section 4.3 and propose heuristics to partially address it. Nonetheless, it is plausible that this issue may persist despite these mitigating measures. The experimental results also reflect this, as the use of a neural network with just one hidden layer took a considerable amount of time to train, surpassing 14,000 seconds. This highlights a critical scalability concern that may limit the practical applicability of the proposed method.

- **Limited experiments**: The validation of the proposed method exclusively on a single dataset may be insufficient to establish its robustness and generalizability. It is advisable to broaden the experimental scope by evaluating the approach across multiple datasets. This would provide a more comprehensive understanding of its performance under diverse conditions and enhance the overall confidence in the proposed methodology. Expanding the experimental validation to encompass a wider range of scenarios would strengthen the empirical foundation of the study.

**Questions:**

The paper refers to Appendix A to detailed explanation. However, Appendix A lacks such a description. I understand that the derivation follows from Elmachtoub & Grigas (2022) but still, it would be nice to incorporate the detailed derivation in the paper. This is also true for eq. 15.

---

> ### Author Response · Authors · 2023-11-23
> **Response to weaknesses and question**
>
> Thank you for the diligent and fair review. We hope that our replies to the listed weaknesses and question below can convince you of the added value of our paper.
>
> W1: Although the proposed method builds upon the SPO+ reformulation framework as proposed by Elmachtoub & Grigas (2022), we believe that the use of a tailored interior point method to derive intermediate results that may generalize better on unseen data is a non-trivial step. Indeed, this contrasts with the prevailing emphasis in the optimization community on optimizing for speed in finding an optimum solution. Additionally, the extension to non-linear forecasters and the mini-batch approach that was added in the revised version of the manuscript substantially enhance the versatility of this reformulation approach.
>
> W2: See remark on ‘Scalability’ in our comment to all reviewers.
>
> W3: See remark on ‘Amount of experiments’ in our comment to all reviewers.
>
> Q1: In the revised manuscript we have extended appendix A to include more details on the reformulation of both the linear and non-linear ERM.

---

> > ### Comment · Reviewer_G7zN · 2023-12-05
> >
> > I'd like to thank the authors for their detailed response! However, I still believe that having more benchmarks and baselines (as indicated by @XyMG) is critical to verify the proposed method. The given experiment is too specific (although practical) and it is hard to assess the comparison. Moreover, it will be helpful if author can specify the scale of experiments in-detail: feature dimensionality, number of variables, number of constraints, etc... I wasn't able to find such details in the paper.
> >
> > I'm glad to see the reduction in runtime brought by using mini-batch and warm start. However, I'd say it is still prohibitory large (~8000 seconds) given that other baselines can do it in ~500 sec.
> >
> > Furthermore, I'm still not convinced that the approach proposes a significantly novel approach w.r.t. Elmachtoub & Grigas (2022).
> >
> > For these reasons, I will maintain my original score of a reject.

---

### Official Review · Reviewer_XyMG · 2023-10-31

**Soundness:** 2 fair
**Presentation:** 3 good
**Contribution:** 2 fair
**Rating:** 3
**Confidence:** 4

**Summary:**

The authors propose an approach for decision focused learning in a setting where a neural network is trained to predict latent objective coefficients for a linear program. The proposed approach builds on previous work that trains a linear model by solving an optimization problem that identifies the best linear model leading to high-quality downstream decisions as evaluated by the true objective coefficients. The authors extend this work by proposing a method for finding the best neural network parameters that lead to high quality downstream decisions. The proposed approach formulates this end-to-end pipeline as an optimization problem which is solved using an interior point method which maintains a current solution, comprised of neural network weights, and iteratively updates the solution to trade off avoiding constraint violation versus rewarding higher-quality solutions. The authors compare three versions of their approach for training both neural networks as well as linear models against three previous approaches: an implicit differentiation approach with quadratic smoothing, a subgradient approach, and a subgradient approach with reformulation. They evaluate these methods on one real world dataset for optimal scheduling of energy storage. The results demonstrate that their approach improves over the baselines in performance while it does take extensive time to train.

With the main strength being the novelty of the approach, there are several limitations in the method and empirical evaluation. If these are addressed, I am happy to increase my score.

**Strengths:**

The main strength of this approach is that their formulation and solving approach are novel, and they demonstrate improved performance on a real world setting over reasonable baselines.

The solving approach of interior point method is promising in that it can potentially combine the gradient-based methods that can be used to solved linear programs with the gradient-based methods for training neural networks. Proper combination of these two has the potential to tightly integrate the learning and optimization components for improved performance as suggested in this work.

The method additionally does seem to give improved performance over the investigated baselines in a realistic setting. Additionally, the contribution of this new setting to the space of decision-focused learning will greatly improve the space by introducing another method for evaluation that has real world impact.

**Weaknesses:**

The main weaknesses of the proposed approach are the running time and the empirical evaluation.

The approach overall seems to take longer due to requiring the solving of a large optimization model with the interior point method, an approach that is known to not scale well. The authors hint that this might be alleviated by using minibatches which seems reasonable in that they could simply iterate by optimizing over a minibatch of problem instances from one iteration to the next. It would be great to understand whether using minibatches improves or harms the training performance as it may make the training process more unstable.

In the evaluation, there are a number of aspects that would improve the paper.

It would be helpful to evaluate the proposed approach against the relevant baselines. For instance, it would be helpful to compare against the other interior point method for decision-focused learning, in the cited Mandi & Guns 2020 paper. Additionally, consider evaluating against the implicit MLE paper [1], differentiable perturbed optimizers [2], dfl without optimization [3], and using CvxpyLayers [4].

Furthermore, it would be helpful to evaluate some of the LP-based settings used in previous work. For instance, the bipartite matching setting from the cited Wilder et al. 2018, the warcraft path planning setting from [5], or the shortest path setting from the cited Mandi and Guns 2020 paper. Since the evaluation is solely empirical, it would help to further improve the empirical evaluation by demonstrating that the method works in more settings.



[1] Niepert, Mathias, Pasquale Minervini, and Luca Franceschi. "Implicit MLE: backpropagating through discrete exponential family distributions." NeurIPS (2021)

[2] Berthet, Q., Blondel, M., Teboul, O., Cuturi, M., Vert, J. P., & Bach, F. (2020). Learning with differentiable perturbed optimizers. NeurIPS (2020).

[3] Shah, Sanket, et al. "Decision-focused learning without decision-making: Learning locally optimized decision losses." Advances in Neural Information Processing Systems 35 (2022): 1320-1332.

[4] Agrawal, Akshay, et al. "Differentiable convex optimization layers." NeurIPS (2019).

[5] Pogančić, Marin Vlastelica, et al. "Differentiation of blackbox combinatorial solvers." ICLR. 2019.

**Questions:**

Is this method potentially applicable to other optimization frameworks which use interior methods for solving? It seems that it requires taking the dual of the downstream optimization problem. Would it be readily applicable for prediction plus optimization for quadratic programs? Is it possible to extend this framework to differentiation of nonlinear optimization problems?

Why is time bolded for the proposed method when it seems to consistently have high running times especially when compared to the implicit differentiation method?


What is the impact of penalizing the difference between the initial predictions in the formulation? It seems that this is present only for the proposed method but not for the baselines whereas the penalty term could be easily added to the implicit differentiation method by adding a penalty term. It might help to preform an ablation study to understand the impact of training using IP versus adding a penalty for deviating from the initial prediction. For instance, this deviation could also be used for the ID models by adding a loss that penalizes deviation from the initial cost prediction.

Along the lines of using a pretrained model, does the cold start method have access to the predictions of the pretrained model as it is solving 15? Do the other methods have access to the pretrained model as well in the cold start?

---

> ### Author Response · Authors · 2023-11-23
> **Response to weaknesses and questions**
>
> Thank you for the diligent and fair review, and the useful suggestions. We hope that our replies to the listed weaknesses and questions below can convince you of the added value of our paper.
>
> W1 (scalabiliyt): See remark on ‘Scalability’ in our comment to all reviewers.
>
> W2 (benchmark methods): We agree with the reviewer that a thorough benchmarking exercise is needed to fully validate the proposed method. In the context of this paper being a proof of concept of a largely unexplored method, we do believe that the current set of methods gives a good first view of the relative performance. The main objective is to show that the method outperforms the available SPO+-based methods (being the subgradient method, Sp-SG, and the current state-of-the-art reformulation without validation performance tracking, Sp-R). Many other decision-focused methods have been proposed and providing a fair comparison requires a diligent effort in ensuring the method is implemented and tuned correctly, which we believe is not feasible from a practical point of view for all the methods proposed. For that reason, we have focused on the method of implicit differentiation, which seems to be the main benchmark from the literature. We did, however, extend our analysis in this revised version to also include the interior point method proposed in (Mandi & Guns, 2020) by considering the log-barrier smoothing function. The result of this method is added under the name ID-LB to Table 1 and Table 3 of the revised manuscript. The results show a slight improvement compared to the ID-Q method, but still a significantly worse performance than what we propose. Please note that the implicit differentiation method with quadratic smoothing, ID-Q, that was also included in the first version is also implemented with Cvxpylayers.  We detail this in Appendix D1 of the revised manuscript.
>
> W3 (limited experiments): Thank you for the useful suggestions of possible other experiments to test the methodology on. See the remark on ‘Amount of experiments’ for our general comment on this.
>
> Q1: The proposed methodology is tailored for linear optimization problems that need to be informed by unknown cost/price information, and is not immediately appliable to quadratic or general non-linear optimization problems. The reason is the requirement of the surrogate SPO+ loss function: to be able to perform the reformulation procedure outlined in Appendix A (which is extended in the revised manuscript), you need a surrogate loss function that can be integrated in the outer optimization problem of training the forecaster. One quality of the SPO+ loss function that is crucial for this is that it involves the objective value of a linear optimization program (as opposed to arguments leading to the optimal value in the case of the regret loss function). Whereas (Elmachtoub & Grigas, 2022) show that the SPO+ loss function is Fisher consistent with the regret loss function in problem setting (1) of our manuscript, it is unclear if a similar surrogate exists for a quadratic (or any general non-linear) optimization problem, and what it would look like. However, similar to the knapsack experiment performed in (Mandi & Guns, 2020), this approach could be implemented for mixed-integer linear problems by considering its continuous relaxation.
>
> Q2: Thank you for pointing this out. We have adjusted the bolding in Tables 1 and 3 of the revised manuscript
>
> Q3: Thank you for this useful suggestion. Due to the limited time for this revision, we were unable to implement this at this point.
>
> Q4: For all models, the following holds:
>
> •	Cold start: the initial price forecast of the MSE-trained model is included as a feature
>
> •	Warm start: has the same features as the cold start, but the starting point of the training constitutes a forecaster that either produces exactly the initial forecast (for the linear re-forecast) or produces prices very close to the initial forecast (for the softplus re-forecaster). The latter is accomplished by pre-training the re-forecaster to minimize the MSE between its output and the initial forecast

---

### Official Review · Reviewer_pKJU · 2023-11-01

**Soundness:** 3 good
**Presentation:** 3 good
**Contribution:** 3 good
**Rating:** 6
**Confidence:** 3

**Summary:**

The paper considers the problem of learning cost functions of linear programs. The decision focused learning aspect incorporates the downstream decisions obtained on solving the estimated linear program. This is usually accomplished by adding a regret loss term in the training procedure. The paper builds upon the SPO+ framework which constructs a convex surrogate for the generally non-convex decision focused loss term. The key idea is to use an interior point method for solving the same surrogate while also incorporating early stopping. The proposed approach extends to both linear and non-linear model classes. Experiments are performed on day-ahead scheduling problem for energy storage.

**Strengths:**

- The paper does a really good job of introducing the literature to a reader not well-versed with this literature. I think section 3 is a really good setup for understanding both the problem and solution space.

- Although the idea mostly builds upon SPO+ framework, I think it is still valuable as it allows non-linear model classes not possible with the earlier approach.

- The proposed approach does well on an important real-world application related to electricity market.

**Weaknesses:**

- I think the comment about treating intermediate points on the path of a interior point solver as intermediate solutions require more justification. I am referring to "However, we argue that when the optimization program is an ERM for training a forecaster, the points on the central path should be regarded as actual intermediate solutions to be tested on the validation set." Please provide some principled justification as this is critical to the early stopping procedure.

- Does the method's extension to neural networks in Section 4.2 work for any general activation function or is it restricted to just the ReLU function? How does the choice of activation function (i.e. 3rd constraint in (15)) affect the optimization problem?

- If possible, can you please add error bars to the result in Table 1.

**Questions:**

Please see weaknesses section above.

---

> ### Author Response · Authors · 2023-11-23
> **Response to weaknesses**
>
> Thank you for the the time and effort spent on this review. Please find below our response to the listed weaknesses.
>
> W1: See remark on ‘Interpretation of interior points’ in our comment to all reviewers.
>
> W2: In our implementation, we used the softplus activation function, which is a smooth variant of the ReLU. This approach would work for any activation function. However, we observe that some activation functions are more suitable for the interior point method in terms of required train time. For example, replacing the softplus activation with ReLU leads to an average increase in train time of around 30% in our experiment. We agree that further investigations are worthwhile to fully understand the interplay between the IP solver and the architecture of the NN, but in the context of this paper, what matters is the generic nature of the proposed framework, paving the way for future improvements.
>
> W3: We agree with the relevance of such sensitivity analysis. However, given the time constraints for this revision, we were unable at this point to add such error bars.

---

### Official Review · Reviewer_ysWa · 2023-11-02

**Soundness:** 3 good
**Presentation:** 3 good
**Contribution:** 3 good
**Rating:** 6
**Confidence:** 4

**Summary:**

Machine learning forecasts are often used as part of downstream decision making tasks, motivating the area of decision-focused learning. In decision-focused learning, the downstream optimization problem is included in the forecasting pipeline by using a task-aware loss function, such as the regret loss. However, these loss functions often have ill-defined gradients, making it difficult to apply gradient descent to minimize the risk. As one way of approaching this problem, a previous work has used a subgradient methods to minimize a convex surrogate of the regret loss. This work proposes an interior point method to solve the forecaster training problem that avoids overfitting by tracking the validation regret obtains by different iterates. An advantage of this approach is that it can accomodate training of neural network forecasters.

**Strengths:**

1. The problem that the authors study--broadly, how to solve optimization problems by machine learning forecasters, is an important and broadly applicable problem. Practitioners from economics and operations research often interested in problems of this flavor, and it's great that this work aims to develop algorithms that can incorporate flexible forecasters to solve optimization problems.

2. The authors do a good job of summarizing and contextualizing previous work, and they carefully distill the shortcomings of previous approaches.

3. The idea of using a "validation performance tracking procedure," inspired by early stopping, is a simple data-driven approach for preventing overfitting. A common concern in solving data-driven constrained optimization problems is that the solution to the problem may not generalize well out-of-sample. I think this procedure offers a simple, transparent, and promising approach of preventing over-fitting. It's an idea that has come to mind before but I actually haven't seen it formally written up in any works so far. Although the idea is simple, I think this idea could be very useful strategy for handling overfitting in constrained optimization problems, and I would like to see this idea developed with more careful theory in future works.

Overall, I recommend to accept this paper and am willing to raise my score if the authors adequately address the questions listed below. This paper tackles an important and challenging problem, and the ideas that they propose (1) a validation performance tracking strategy to evaluate iterates of an interior point method (2) using interior-point methods to facilitate using neural network forecasters as part of optimization problems--are promising and have the potential to be useful in practice.

**Weaknesses:**

1. One weakness of this paper is that the authors do not provide very much theoretical justification for why their validation performance tracking procedure may yield improved out-of-sample performance. As a result, the strategy that they propose is largely a heuristic. That being said, I think this heuristic is quite promising and I would hope that future works can develop a rigorous theory to justify this approach.

2. There are some clarity concerns that I had (see questions below), but I believe that these can be addressed with proper writing and motivation.

3. The authors demonstrate a nice proof-of-concept, but I would be interested in seeing a more thorough empirical evaluation, even on synthetic optimization problems.

**Questions:**

1. It is somewhat unclear what "the ERM" is in the following sentence in the introduction ``Secondly, the ERM can be re-written to a single-level
optimization program by applying duality theory`` -- what ERM problem are the authors referring to? The problem of training the forecaster? The problem of minimizing the regret loss? While this is clarified later in the paper, it would be helpful to make this clear in the introduction as well.

2. In equation 4, the set $S$ is not yet defined? I assume that $S$ is the feasible set for $x$. From the example in Equation 1, I presume that $S = \{ x \mid Ax \geq b}.$ I think it would be helpful if the authors could reiterate the downstream optimization task again in Section 3 for clarity.

3. Do the authors have any intuition on why training a neural network to minimize the SPO+ loss function $l^{SPO+}$ performs poorly?

4. How does the validation performance tracking procedure that the authors propose compare to just solving the original SPO+ problem (Equation 9) with regularization?

5. Could the authors elaborate more on the following sentence: ``We argue
that when the optimization program is an ERM for training a forecaster, the points on the central
path should be regarded as actual intermediate solutions to be tested on the validation set"?

6. The authors make the following comment: ``Since the proposed methodology currently does not involve mini-batches, the IP solution method processes all the train data in a single run.``
Do the authors think it would be possible to implement a ``stochastic`` version of the IP method where a new mini-batch of data could be used to solve for each the optimal parameters in each iteration of the interior-point method? Could this also prevent overfitting? Or do the authors expect this approach to be unstable?

7. The idea of warm-starting the forecaster by first fitting the forecaster with the MSE loss is another useful heuristic--this one is probably used in various prior works. Also, it would great if the authors could comment on how much benefit is derived from each stage of training (how far does fitting with the MSE loss get you? how much additional benefit does the IP method provide?). Is the forecaster that is fit with the MSE loss the ``Initial FC`` in Table 1? In addition, the authors comment that a ``refined`` set of features are used to train the second-stage forecaster, how is this refined set of features selected? Also, could the authors add citations to other works that use the warm-start strategy?

8. I am struggling to interpret Figure 2. Could the authors explain the difference betwen the SP-R-IP methods they evaluate in the experiments? What is the expected behavior under each of these 3 different algorithms? Are they all variants on the authors' proposed approach? What is the reason that the validation regret of SP-R-IPs and SP-R-IPd oscillates (instead of decreasing monotonically)?

---

> ### Author Response · Authors · 2023-11-23
> **Answer to questions 1-7**
>
> Thank you for the diligent review. Please find below our answers to the specific questions.
>
> Q1: It is indeed the problem of training the forecaster, in this case with the regret loss function. We have re-formulated this to “Secondly, the training problem can be …” in the revised manuscript.
>
> Q2: We agree that it improves clarity to reiterate the specifics of the set S, which indeed corresponds to the set {x|Ax>=b}. We have included this in the revised version of the manuscript.
>
> Q3: As a clarification, we would like to remind the reviewer that our approach (any form of Sp-R-IP) involves using the SPO+ loss function. The good performance of our model therefore contributes to the validation of the SPO+ loss function, which was already theoretically shown to be consistent with the regret in (Elmachtoub and Grigas, 2022). What we see is that training a model with their originally-proposed subgradient descent method may be less performant than our proposed interior point method. The intuition behind this is two-fold: the first reason is that training a forecaster to minimize such a decision-focused loss function is a highly non-convex problem. Whereas our method smoothens the problem by introducing the log-barrier term, (sub)gradient methods are very prone to getting stuck in local optima. This intuition is underscored by the results in Table 3, where we show that the (sub)gradient methods ID-Q and Sp-SG perform very poorly when cold started, whereas the proposed Sp-R-IP method is able to attain similar performance when cold started compared to the warm start. The second reason is that the expression of the subgradient will result in model updates even when the current forecast leads to the perfect decision, which we showcase in Appendix D.2.
>
> Q4: We observe that when a regularizer is used, the difference between the best performing model along the points accessed on the central path and the final resulting model generally reduces, but there is still a noticeable difference. This can be concluded from Tables 1 and 2: Table 2 shows that for the Sp-R methods, we explore multiple values of the regularization, whereas this clearly still results in improved performance of the Sp-R-IP methods as compared to the Sp-R method in Table 1.
>
> Q5: See remark on ‘Interpretation of interior points’ in our comment to all reviewers.
>
> Q6: See remark on ‘Scalability’ in our comment to all reviewers.
>
> Q7: The row named “Initial FC” indeed represents the performance of the forecaster that was trained to minimize the MSE. The relative performance (second column in Table 1) of the other methods constitutes the difference compared to this baseline. The regret is improved by 19% for the best model, Sp-R-IPs-softplus. The features were selected based on our experience in energy systems. Here, we leverage our understanding of the market clearing mechanism in Europe that considers supply and demand in European countries jointly. The features used in re-forecasting should in this sense capture the most important information.
> As detailed in our general comment, this warm start naturally emerges from the observation that decision-focused learning methods seem to perform poorly when not warm started, even when the original price forecast is included in the feature set. Other examples of such a two-stage methodology are extreme learning machines and large language models adapters, see Section 4.3.

---

> ### Author Response · Authors · 2023-11-23
> **Answer to question 8**
>
> The models compared in these figures are indeed variants of our proposed approach of training the forecaster by solving the ERM that follows from using the SPO+ loss function and reformulating it to a single-level optimization program, solving it with an interior point method. The way they obtain points along the central path differs:
>
> •	Sp-R-IP directly uses the default IPOPT algorithm for getting the values of \mu (the barrier term). This algorithm prioritizes speed of finding a solution over granularity and amount of points obtained along the central path. This makes sense in the general setting where people try to find an optimal solution as fast as possible. However, this results in fewer evaluations of the validation performance as is shown in Figure 4: only 4 values of \mu are considered. Even though this is not necessarily a problem for the performance on the train set, where you expect to find the best result for the smallest value of \mu, this may lead you to oversee manifestations of the forecaster on the central path that generalize better on the validation set, as can be seen in Figure 2b.
>
> •	To overcome that issue, we propose to manually set the value of \mu for which IPOPT should optimize iteratively:
>
> o	Sp-R-IPs (static) decreases $\mu$ at a steady pace, in the case of Figure 2 by a factor 1.5, i.e. $\mu_{n+1} = \mu_{n}/1.5$.
> o	Sp-R-IPd (dynamic) decreases $\mu$ dynamically based on the validation performance, i.e. slowing down the rate of decreasing the value of $\mu$ based on equations (12)-(14).
>
> There is no immediate explanation as to why the validation performance oscillates, other than that there is no perfect match between the train and validation set. This is a behaviour that can be expected in gradient descent approaches too.

---

### Author Response · Authors · 2023-11-23
**General comment to reviewers**

We would like to thank all the reviewers for the time and effort spent on reviewing this paper, and providing us with sensible and useful comments and suggestions. First, we’d like to address some of the recurring comments in this general comment:

1.	Interpretation of interior points

The neural network parameters (weights and biases) are decision variables in the ERM problems (9) and (15). The problem with the original SPO+ reformulation framework is that solving the optimization program immediately yields the optimal solution on the training set. It is well-known in machine learning that such a solution, even with regularization, is likely to overfit the forecaster on the train dataset. In that regard, the interior points obtained when solving that ERM with an interior point solver are the direct equivalent to the intermediate realizations of the forecaster obtained by updating the weights and biases by using the gradient in traditional gradient descent training. Indeed, the intermediate solutions with decreasing value of the barrier term \mu cause the forecaster to get stepwise closer to the optimal solution on the training data. What is intuitive, and what we also show in the results, see e.g. Figure 2, is that within this update procedure, some intermediate realization of the forecaster likely performs better on unseen (validation) data.

2.	Scalability

The fact that we use a second order method (interior point solver) makes the proposed method inherently less scalable than a more common first-order method like gradient descent. It is important to remark here that our results show that the gradient descent-based methods perform very poorly when not warm started, see Table 1. This insight leads to the conclusion that likely, the highly non-convex nature of a decision-focused learning problem causes such gradient descent methods to easily get stuck in a local optimum. For that reason, a two-stage approach where first a forecaster is trained in a generic setting, while in a second stage a re-forecaster is trained to minimize a decision-focused learning loss function, is a natural way to approach this problem. As such, the fact that the decision-focused learning method is not highly scalable is not necessarily a problem since one could implement the re-forecaster through a problem with reduced dimensionality, as we implemented in this paper.
However, we did address the scalability issue in the revised version of the manuscript by adopting a mini-batch approach. Whereas it is not immediately obvious how you would implement a direct equivalent to gradient descent-based minibatches, we adopted a naïve approach where the training dataset is split up in M batches. For all of the batches, the training procedure outlined in Algorithm 1 is executed. This leads to a trade-off between information available in training the model (which is lower for the mini-batches because of the reduced amount of examples) and the variety of forecasters trained (which will be higher than the single forecaster in the single batch approach) that may generalize better on unseen data. In appendix E, we show the sensitivity of the train time and the regret on the validation set when adopting different sizes of mini-batches. The train time per batch drastically decreases with decreasing batch size, while the regret performance is similar to that of the single batch optimization. When such calculations are performed in parallel fashion, the absolute speed-up of the training is significant.
A possible extension of this mini-batch approach that could be considered in future work entails finding a consensus among the minibatches (as opposed to training the mini-batches independently as is currently implemented), e.g. by adopting techniques from federated learning.

3.	Amount of experiments

We agree that it would be interesting to see the performance of our proposed method in other settings. At this stage, we were unable to implement that. However, we do believe that the single experiment showing improved results in comparison to well-established decision-focused benchmark should, in itself, provide a compelling argument for the validity of the model since (i) it involves a real-life case study using actual data and (ii) this is a highly relevant problem that many decision-makers in energy systems face on a daily basis.

---

### Author Response · Authors · 2023-11-23
**Changes in revised manuscript**

The following changes were made to the manuscript to improve/clarify certain aspects:

1.	We have included the heuristic procedure for implementing the proposed method with mini-batches as explained above. This is added to the discussion in Section 4.3. The sensitivity analysis showing how the train time and regret performance vary for varying sizes of mini-batches is added in Appendix E. We have also re-run the results for the Sp-R methods using a batch size of 16. For the models Sp-R-IPs-softplus and Sp-R-IP-softplus this has resulted in an improved regret performance. Tables 1 and 3 were adjusted accordingly.

2.	The method of implicit differentiation through the KKT optimality conditions using a log-barrier smoothing term as was proposed by Mandi & Guns (2020) is added as one of the benchmark methods in Tables 1 and 3. The implementation details are added in Appendix D.1. As a general insight we see that it slightly outperforms the implicit differentiation benchmark with quadratic smoothing, but still significantly underperforms in terms of regret on the test set as compared to our method.

3.	We have (slightly) extended the text in Section 4.1 to further clarify the interpretation of the interior points as intermediate solutions to be checked on the validation set.
4.	To adhere to the 9-page limit, we moved details of the experiment regarding the data used and the initial forecaster to Appendix B. Some aspects of the different decision-focused re-forecasters were moved to appendix C.

5.	We further elaborated on the derivation of ERMs (9) and (15) in Appendix

---

### Meta-Review · Area_Chair_GSSR · 2023-12-09

**Metareview:**

This paper is focused on decision-focused learning and addresses the issue of ill-defined gradients with task-aware loss functions. It proposes an interior point method to solve the forecaster training problem that avoids over-fitting by tracking the validation regret obtains by different iterates.

The reviewers' appreciated the novelty of the approach but also identified weaknesses in both the method and empirical evaluation. The authors' rebuttal has addressed some of the concerns, but the outstanding concern is improving the experimental evaluation as per reviewer comments' (more synthetic and real-world evaluations, and experimenting with different activation functions for neural networks).

Therefore, I recommend rejecting the paper. I strongly encourage the authors' to resubmit after improving the experimental evaluation and positioning the novelty of the proposed approach w.r.t Elmachtoub & Grigas (2022).

**Justification For Why Not Higher Score:**

Very weak experimental evaluation.

**Justification For Why Not Lower Score:**

N/A

---

### Decision · Program_Chairs · 2024-01-16

Reject